# Dynamic Transaction Confirmation Sharding Protocol For Alliance Chain

**Nigang Sun [1], Junlong Li [2],\* and Yuanyi Zhang [2]**

1  School of Microelectronics and Control Engineering, Changzhou University, Changzhou 213000, China; ngsun@cczu.edu.cn
2  School of Computer Science and Artificial Intelligence, Changzhou University, Changzhou 213000, China; revanton@icloud.com
\*  Correspondence: wddygyys@163.com

**Abstract:** Alliance chain has gained widespread popularity in industrial and commercial fields due to its multi-centralization and node manageability. Current implementations of the alliance chain suffer from scalability obstacles, such as communication congestion and throughput drop, when the number of nodes increases. In this paper, a novel dynamic transaction confirmation sharding protocol is proposed, which improves transaction processing efficiency by partitioning nodes and assigning different transactions to different shards. It utilizes dynamic transaction confirmation consensus as a sharding intra-consensus mechanism to minimize message size and package transactions into microblocks, which modifies communication content during transaction propagation among shards and reduces network congestion and shard reconfigure cost. The protocol leverages a review system and reputation model to identify and punish malicious nodes and also incorporates a verifiable random function for node configuration, which ensures a sufficient number of honest nodes within the shard and prevents repeated consensus processes. Simulation results show that the proposed protocol outperforms mainstream used permissioned chain sharding protocols Attested HyperLedger and Sharper, achieving a throughput improvement of at least 20%. This protocol is suitable for scenarios requiring high throughput and reliability in industrial and commercial fields such as finance, logistics, and supply chain management. Even if the number of alliance chain nodes increases to the usual maximum, or there are some faulty nodes, the protocol can still maintain stable performance.

**Keywords:** blockchain; alliance chain; scalability; blockchain scaling; sharding protocol

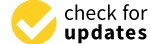



## 1. Introduction

Alliance chain is a type of blockchain that provides decentralization and node manageability, making it widely used in industries including finance, supply chain management, and healthcare [1]. However, as the number of nodes increases, communication congestion and throughput drop become significant scalability obstacles for alliance chain [2]. This situation arises because all transactions within the blockchain structure require nodes to utilize their computational and storage resources [3,4], resulting in substantial wastage of time and space [5]. The manifestation is the bottleneck of low throughput and high latency caused by the blockchain's difficulty bearing the cost of consensus process and ledger storage [6]. Researchers have suggested various strategies to tackle performance issues and fulfill the practical requirement of enhancing blockchain resources to manage the rising volume of transactions [7]. Existing solutions to achieve blockchain scaling are classified into two categories: off-chain and on-chain [8]. Off-chain solutions adopt a layered system to simplify the blockchain architecture by aggregating the transfer of resources generated by fine-grained payments managed separately in multiple asynchronous subsystems. Only the net result is stored in the blockchain, thus avoiding the high computational cost of traditional blockchain systems [9]. In practical applications, the off-chain payment network

requires frequent opening and closing channels to achieve consensus on all completed transactions on the blockchain [10,11]. This process appreciably impacts the throughput of the blockchain, thus limiting the advantages provided by off-chain solutions [12,13]. On-chain solutions enhance the functionality and data processing capabilities of the blockchain by improving its protocols and mechanisms [14]. Increasing the block size is typical of on-chain solutions, as it allows for higher transfer limits and reduces costs associated with transfers compared to traditional methods [8]. This approach can have negative implications for block propagation efficiency in terms of time and increase the risk of blockchain forks, leading to a higher probability of orphan blocks and increased maintenance costs [15]. The inefficiency of consensus protocols is the primary cause of blockchain scalability issues [16]. The research community has made significant efforts to address this problem by exploring various innovative consensus methods, yielding remarkable results [17–19]. There is no need to coordinate or manage various subsystems, and the recording, verifying, and retaining all transactions are characteristics of on-chain solutions. These features make on-chain solutions a mainstream choice for scaling blockchain in environments with varying network bandwidth and computing resources [20].

Sharding protocol is commonly used in distributed databases and cloud infrastructures, which can divide an enormous database into small data fragments and store these fragments on different servers for fast and efficient data management [21]. Elastico [22] pioneered the combination of the sharding protocol with the blockchain in 2016, avoiding the mandatory duplication of communication and computing overhead for each participating node. The sharding protocol has been comprehensively researched and verified in academic papers [23–29], confirming its effectiveness in enhancing throughput, reducing costs, and preserving decentralization. It has emerged as a prevailing solution for on-chain scaling [30]. During the initial stage of the sharding protocol's development in blockchain, it effectively addressed the resource-intensive challenges public blockchain face [31]. As alliance chain encountered scalability challenges similar to public blockchain in the use process [32], researchers applied sharding protocol to alliance chain, offering valuable assistance in achieving efficient transaction processing [33]. The Atomix protocol and ByzCoinX in OmniLedger [34] enhance cross-shard and intra-shard communication and resource management. Nevertheless, client dependency of OmniLedger cause communication overhead to become the limiting factor [29]. RapidChain [35] utilizes a lightweight reconstruction protocol and reduces the data transmitted in each transaction, which mitigates the bottleneck issue of transaction communication overhead in early sharding protocols. Dang proposed AHL (Attested HyperLedger) [36] to promote the sharding protocol to applications in permissioned environments beyond cryptocurrencies. The optimization of BFT (Byzantine Fault Tolerance) consensus in AHL reduces the maximum number of nodes required for a single shard to improve the throughput of large-scale permissioned chains, but the actual application is limited by insufficient scalability and an unbalanced workload [37]. Amiri proposes Sharper [7], which uses a hash-based sharding strategy and a BFT-based algorithm to ensure fast transaction distribution processing and process cross-shard transactions through non-overlapping committees in parallel computing, showing excellent scalability and load balancing. Sharper still faces challenges in the complexity of node election and consensus algorithms and issues related to data access efficiency [38]. FleetChain [39] utilizes FBFT (Fast byzantine fault tolerance) to improve communication and processing efficiency and employs RSTP (Responsive sharded transaction processing) for improved cross-shard consensus via multi-signature aggregation. Sharding protocol has improved the scalability of the alliance chain, but there are still insufficient performance problems in transaction verification and consensus processing. As the number of members increases, the communication bandwidth and time required for broadcasting and message collection consume a significant amount of resources, thereby reducing protocol efficiency.

This paper proposes a dynamic sharding protocol for transaction confirmation. During the shard configuration phase, a random function is utilized to elect a Master node to preside over the epoch, ensuring the impartiality of the consensus process. Subsequently,

each node is randomly assigned to a specific committee based on its identity information. The protocol employs a dynamic transaction confirmation consensus mechanism, which enhances the efficiency of consensus within the shard, appreciably improving the performance of the alliance chain system. Once consensus within the shard is achieved, the Leader node encapsulates the microblocks to reduce communication overhead. The protocol includes a review mechanism and a reputation model to constrain node behavior. In general, the protocol markedly improves the performance and scalability of the alliance chain.

Simulation results show that in the alliance chain with the same number of nodes, the proposed protocol increases the transaction throughput by 20% compared with the current mainstream permissioned chain sharding protocol and exceeds the high-performance Fleetchain by 19%. As the network scale expands, the alliance chain system based on the protocol maintains a constant performance advantage.

## 2. Related Concepts

### 2.1. Alliance Chain

Alliance chain is a permissioned blockchain network involving authorized entities collaborating to manage and maintain data and transactions [40]. Unlike public blockchain, participants in alliance chain establish a cooperative relationship to make decisions and manage the chain's operations collectively.

The primary purpose of alliance chain is to create a trusted collaboration platform within a specific industry or organization. It can be utilized for various purposes, such as payment and settlement between financial institutions, supply chain management, sharing of medical records, and data exchange between government agencies [41]. By providing decentralized, transparent, traceable, and secure transaction records, alliance chain enhances trust among participants and enables efficient data sharing and automation of business processes [42].

While the security of an alliance chain is relatively high due to the authorized entities involved, there is still a degree of centralization risk compared to the fully decentralized public blockchain [43]. Additionally, the scalability of an alliance chain is restricted due to the limited number of participants and transaction volume, resulting in lower throughput when compared to public blockchain [44]. Furthermore, the governance and consensus mechanisms of alliance chain require participants to reach a consensus, which can lead to a slower and more complex decision-making process [45]. Despite these limitations, alliance chain remains a valuable blockchain solution in specific cooperation scenarios. It offers trusted data sharing and efficient management of business processes, making it suitable for industries and organizations that prioritize security, collaboration, and data integrity [45].

### 2.2. Practical Byzantine Fault Tolerance

PBFT(Practical Byzantine Fault Tolerance) algorithm, proposed by Castro in 1999, has a communication complexity of $O(n^2)$ ($n$ refers to the size of the data scale, which is related to the number of nodes participating in the algorithm) and is used to build Byzantine fault-tolerant distributed systems. The PBFT algorithm is implemented by two types of nodes: master node and consensus node. The identity of the master node (denoted by $p$) is determined by the view number ($v$) and the number of nodes ($R$). The following Equation represents the election mechanism for the master node.

$$p = v \bmod R \tag{1}$$

The PBFT consensus process includes the following steps: request, pre-prepare, prepare, commit, and reply. After receiving a request message from a client, the master node creates a pre-prepare message and sends it to all consensus nodes. Upon receiving and validating the pre-prepare message, the consensus nodes send prepare messages to all nodes. When a consensus node receives more than 2f + 1 (f is the number of Byzantine fault nodes in the system) valid prepare messages from non-self nodes, it sends a commit message to other nodes. When a node receives 2f + 1 valid commit messages from different

nodes, it aggregates all the commit messages and sends a reply to the client. The consensus process concludes when the client receives reply messages from f + 1 different nodes.

Most instant sharding blockchains use PBFT or its variants as their intra-shard consensus protocol [20]. These variants [23,27,34] adjust the validation content and propagation method to increase tolerance to node inflation and Byzantine failures.

### 2.3. Sharding Protocol

Network sharding, transaction sharding, and state sharding are the state-of-the-art mechanisms that perform blockchain sharding protocol in the modern world [46]. Network sharding divides the entire blockchain network into several shards so that different shards can process part of the transactions in the entire blockchain at the same time. Transaction sharding assigns transactions to different shards and allows them to be executed concurrently. State sharding separates and saves the entire ledger in shards, which can reduce the storage burden of network nodes.

Sharding protocol can be decomposed into the following stages: shard configuration, intra-shard consensus, cross-shard protocol, and reconfiguration [31]. The shard configuration stage determines the shards to which the nodes belong and the transactions that each shard will process [22,34,36]. After completing the previous step, the validator nodes in the same shard pass messages according to the internal consensus protocol to reach a consensus on the entire shard. Researchers divide intra-shard consensus into two types: PoW-based [28,29] and BFT-based [7,22,34,35]. The cross-shard protocol uses transaction-related shards as the basic unit for processing cross-shard transactions to generate blocks that contain cross-shard transaction state transitions [34,35]. The reassignment step shuts down validator nodes after one shard epoch and swaps to other shards to maintain the integrity of each shard and avoid attacks from adversaries that slowly adapt [22,35].

Although experimental setups or methodologies for validating different technologies may vary, throughput and latency remain common metrics for evaluating protocol performance [31].

## 3. Dynamic Transaction Confirmation Sharding Protocol

The protocol adopts a dynamic transaction confirmation algorithm to achieve consensus on transactions within shards. It utilizes microblocks to transfer transaction information between shards, thereby directly improving the transaction verification efficiency of the alliance chain. Additionally, random node selection and shard configuration are implemented along with a review system and reputation model to drive nodes to maintain the validity of consensus processes.

### 3.1. Network Infrastructure

The protocol design incorporates three types of validator nodes: consensus, Leader, and Master nodes. Each subcommittee comprises all consensus nodes and Leader nodes in a single shard, and the only consensus committee comprises all Leader nodes and Master nodes. Consensus nodes validate transactions and submit the final consensus result to the Leader node within their respective subcommittee. The Leader node packages the transactions completed in consensus into microblock and submits microblock to the unique Master node for each epoch. The Master node organizes the validation of microblocks within the consensus committee and packages the validated microblocks for upload to the alliance chain. After the status of all validator nodes is synchronized, each Leader node will randomly generate a string. The Master node merges this set of strings, adds the latest block hash, and finally performs a secure hash algorithm(i.e., SHA256) calculation on the result to get epochRandomness. The Master node broadcasts the obtained epochRandomness to the whole network, and then the system executes the view replacement protocol to transition to the next epoch.

At the beginning of the epoch, all nodes combine the (*IP*, *PK*) identity information group (representing their internet protocol address and public key respectively) and the

epochRandomness sent by the system to calculate their identity *ID*. The calculation method is provided in Equation (2).

$$ID = SHA256(IP + PK + epochRandomness) \tag{2}$$

Each node in the system executes a modulo operation using its *ID* and the total number of shards. The result of this modulo operation corresponds to the number of the committee to which the node belongs (in the range of [0, 1, 2, ..., (total number of shards)-1]). In each subcommittee, the node with the smallest ID will become the Leader node. Each Leader node uses the private key (hereinafter referred to as SK) as input to execute VRF (verifiable random function), judges whether it is the Master node according to the output result, and broadcasts verifiable selection information contained in the result if it judges itself as the Master node. Node state transitions are depicted in Figure 1. Algorithm 1 describes the node assignment and selection algorithm step-by-step using pseudocode.

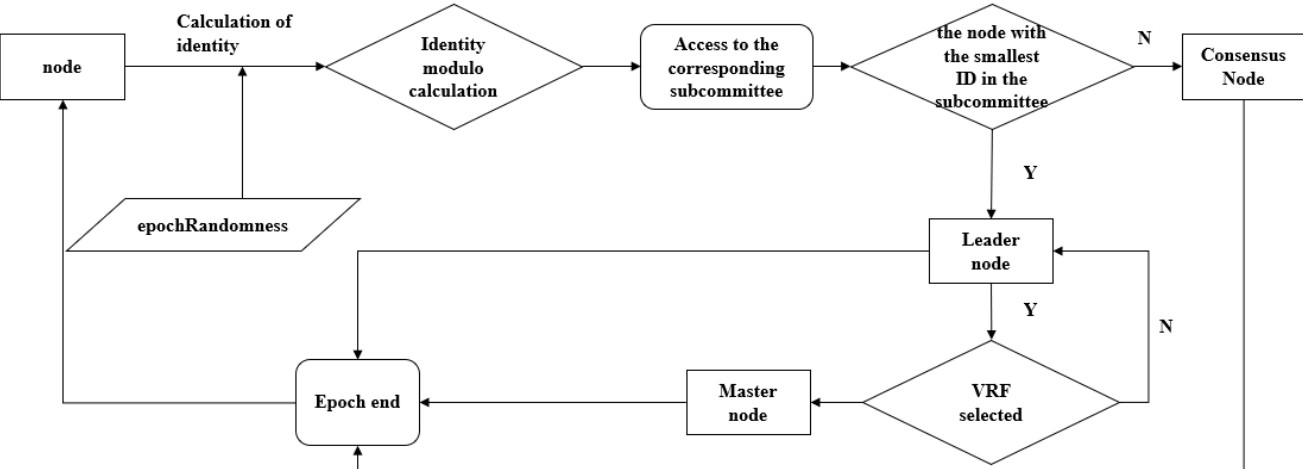

**Figure 1.** State transition of node. A node obtains an ID and determines its shard membership by performing a modulo operation. The node enters the shard and actively participates in the subsequent selection process for selecting the Leader node. If a node becomes a Leader node, it also participates in the selection process for selecting the Master node.

It is crucial to record node information in the shard during the implementation process. This information is directly linked to subsequent node selection and message sending. Figure 2 shows the key codes.

```cpp
void Committee::ShuffleNode(std::unique_ptr<Node> & node) {

    for (auto it:GetCommitteeMembers()){ //
        if (node->GetNodeAdd() == it){
            for (auto  iter:GetCommitteeMembers()){
                node->committe_seq = sequence;
                if (node->GetNodeAdd() != iter){
                    node->_otherCommitteeNodes.emplace_back(iter);
                }
            }
        }
    }
}
```

**Figure 2.** The C++ codes for retrieving the shard and other node information after the node enters the shard. Subsequent node communication relies on this information.

---

**Algorithm 1** Node assignment and selection

---

**Input:** *IP, PK, SK*
**Output:** *Node_ID, Node_shard, Node_state*
  1: *k* = total number of shards
  2: *Shard_number* $\in [0, k-1]$
  3: Identity computing:
  4: Node receives the *epochRandomness* broadcast
  5: *Node_ID* = *SHA256(IP + PK + epochRandomness)*
  6: Node assignment:
  7: *Node_shard* = *Node_ID mod k*
  8: Node selection:
  9: **if** *Node_ID* is the smallest in the shard **then**
 10:   *result = VRF(SK)*
 11:   **if** *rusult = yes* **then**
 12:     *Node_state* = Master node
 13:     broadcast *result*
 14:   **else**
 15:     *Node_state* = Leader node
 16:   **end if**
 17: **else**
 18:   *Node_state* = consensus node
 19: **end if**
 20: **return** *Node_ID, Node_shard, Node_state*

---

### 3.2. Transaction Consensus and Review Mechanism

The client broadcasts the transaction request message to the entire nodes. The remainder *r* obtained by the node according to the hash value of the transaction modulo *k* (the number of shards) is the serial number of the shard that processes the transaction. The Leader node sends the transaction confirmation threshold (hereinafter referred to as *Tct*) to the consensus nodes in the shard.

The consensus node verifies the transaction information, adds the correct transaction to the transaction pool, and broadcasts a confirmation message. After the Leader node receives *Tct* confirmation messages, it queues the transaction into the encapsulation queue. When the epoch time is reached, the Leader node will encapsulate the transactions in the queue into a microblock and send the microblock information to the Master node. Figure 3 depicts the intra-shard consensus.

After receiving a microblock, the Master node will verify the microblock and its internal transactions with all Leader nodes using the PBFT algorithm. On the premise that the consensus of *k* microblocks is completed, the whole network nodes adopt the bigblock synchronization state composed of all microblocks. Algorithm 2 uses pseudocode to describe the consensus steps of transactions in the subcommittee and consensus committee.

The validator nodes receive a bigblock sent by the Master node and compare the content of the bigblock with the already verified transactions through their local transaction pool. If a validator node discovers a transaction submitted by a Leader node but has not been verified, the validator node will send a challenge message (including the microblock information to be reviewed and the position of the transaction to be reviewed ) to the Master node. The Master node forwards the received challenge message to the other $k-1$ shards (excluding the initially processed shard) for verification. The Master node reaches a consensus on the verification results of the $k-1$ shards in the consensus committee. If more than $\frac{2k}{3}$ of the shards in the verification result consider the transaction incorrect, the Master node packages the challenge message and review result into a block and uploads the block to the alliance chain. During the implementation process, it is crucial for nodes to monitor the messages and blocks within the alliance chain network. This function plays a critical role in transaction consensus and on-chain blockchain operations. Figure 4 shows these key codes.

---

**Algorithm 2** Transaction consensus

---

**Input:** *transaction*
**Output:** *bigblock*
 1: consensus nodes ← Leader node sends *Tct*
 2: **if** consensus node is not to receive ← *Tct* message **then**
 3:    broadcast *Tct*-no message
 4:    **while** *Tct*-no messages >2 **do**
 5:       Replace the Leader node
 6:       Leader node sends *Tct*
 7:       Node that has not received *Tct* broadcast *Tct*-no message
 8:    **end while**
 9: **end if**
10: Client sends *transaction*
11: transaction assignment:
12: $r = Hash(transaction)\ mod\ k$
13: the $r$ shard ←*transaction*
14: nodes within the shard receive and verify *transaction*
15: **if** *verification result = true* **then**
16:    forward *confirmation* message
17:    transaction pool ← *transaction*
18: **end if**
19: **if** *Leadernode = true* **then**
20:    **if** *confirmation*>=*Tct* **then**
21:       transaction queue← *confirmation* messages and *transaction*
22:       transaction pool ← *transaction*
23:    **end if**
24:    **if** end of epoch time **then**
25:       *microblock*← transaction queue
26:       Master node← *microblock*
27:    **end if**
28: **end if**
29: Leader nodes← Master node broadcasts *microblock*
30: **if** Leader node receives *microblock* **then**
31:    Leader nodes← *Prepare* massage
32: **end if**
33: **if** *Prepare*>=$\frac{2k}{3}$ **then**
34:    Leader nodes← *Commit* massage
35: **end if**
36: **if** *Commit*>=$\frac{2k}{3}+1$ **then**
37:    *bigblock*← *microblock*
38:    **if** *num of microblock = k* **then**
39:       all nodes← *bigblock*
40:    **end if**
41: **end if**
42: **return** *bigblock*

---

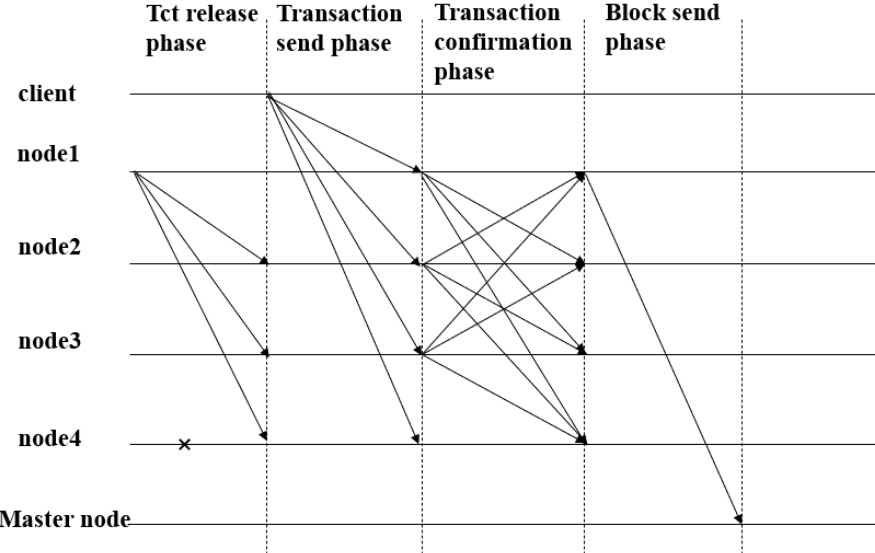

**Node 4 with the " ✕ " is a failed node that cannot receive or send messages.**

**Figure 3.** Flow chart of dynamic transaction confirmation consensus mechanism. There are four consensus stages: *Tct* release, transaction send, transaction confirmation, the Leader node sends microblocks to the Master node.

```cpp
NetworkNode::NetworkNode() :_nAddress(Network::instance().AssignAddress()) {
    std::thread([this]() {
        while (true) {
            try {
                auto msg = Network::instance().RecvMsg(GetNodeAddress());
                OnRecvMsg(msg.src, msg.msg);
            } catch (...) {
                std::this_thread::sleep_for(1s);
            }
        }
    }).detach();
}
```

**Figure 4.** The C++ codes for multiple nodes monitor the network simultaneously, querying whether messages and blocks are sent to themselves in the network.

### 3.3. Reputation Model

In the PBFT-based intra-shard consensus, the consistency of the alliance chain is ensured through the two state synchronizations. The intra-shard consensus of the dynamic transaction confirmation sharding protocol only performs state synchronization once. Therefore, in addition to the confirmation mechanism, a reputation model must be used to force nodes to maintain system security jointly and distinguish malicious nodes from normal nodes. Alliance chain nodes can have a one-to-one mapping relationship with enterprise entities, so economic games can be used to motivate all nodes to maintain system security. Reputation is obtained by recording and calculating behavioral information, which is used to evaluate the reliability of nodes and verify their right to speak. The reputation incentive and penalty model is shown in Figure 5.

In order to increase the cost of malicious behavior, a node malfeasance counter is used in the model. Nodes are assigned a reputation score of $S$ when they join the system, which will be subject to changes and retained throughout subsequent activities. When a node actively maintains system security, it receives an incentive reputation score of $P$. Conversely, if a node's malicious behavior is detected, its reputation score will be deducted

by an amount represented as *Q*. The relationship between reputation *S*, incentive cases *P*, deduction cases *Q*, and the number of misoperations *T* is shown in Equation (3).

$$S = S + P - QT \qquad (3)$$

For example, a Leader node has an initial reputation score *S* of 100 (nodes below this reputation score will be removed from the network). In each epoch, completing a normal transaction consensus will get an incentive reputation score *P* of 100, while an incorrect consensus and sending a transaction whose verification does not meet the *Tct* (Hereinafter referred to as failed transaction) will deduct a reputation score *Q* of 50 and 30, respectively.

In the first epoch, the node completes the consensus, $T = 0$, $S = 100 + 100 = 200$.

In the second epoch, the node completed the consensus but sent a failed transaction, $T = 1$, $S = 200 + 100 - 30 * 1 = 270$.

In the third epoch, the node had a incorrect consensus, $T = 2$, $S = 270 - 50 * 2 = 170$, and sent failed transaction, $T = 3$, $S = 170 - 30 * 3 = 80 < 100$. Therefore, the node is removed from the network. The amount of reputation lost for doing evil is proportional to the number of times doing evil. When a node's reputation drops below a specified value, the node will be removed from the network.

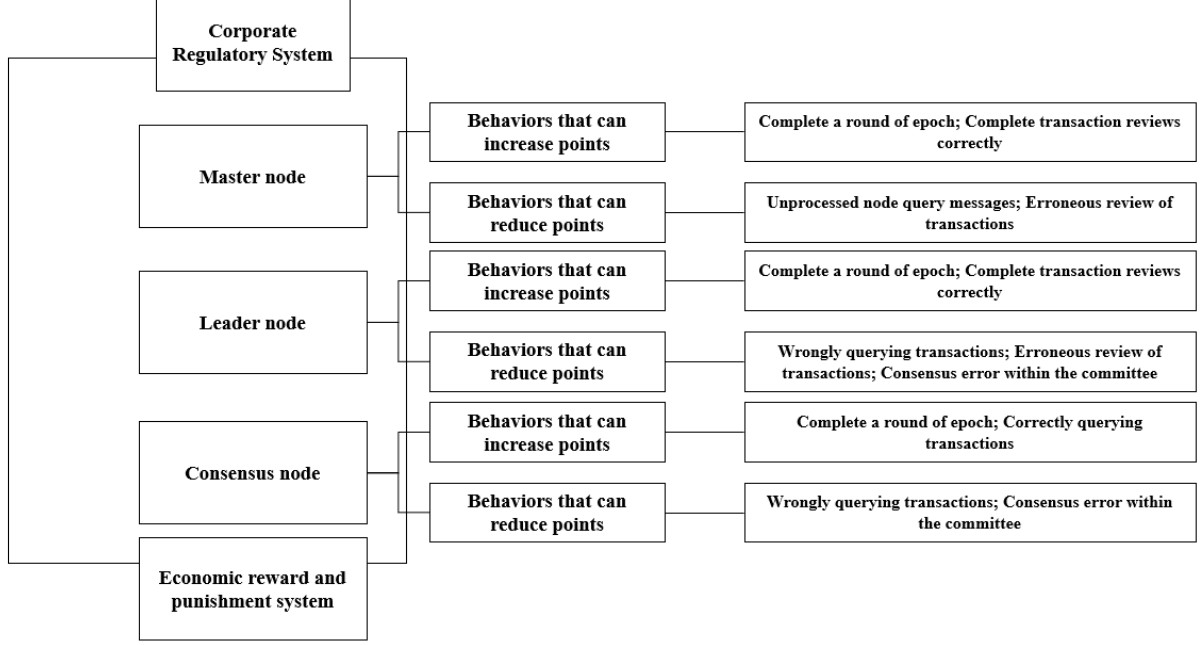

**Figure 5.** Reputation incentive and punishment model. The different types of nodes exhibit variations in incentives and penalties, and they can be integrated with enterprise management and economic systems.

### 3.4. Adjustment Of Transaction Confirmation Threshold

In the intra-shard transaction consensus stage, the transactions confirmed by *Tct* consensus nodes will be encapsulated into microblocks and delivered to the consensus committee. Therefore, the time and security of transaction confirmation are directly affected by *Tct*. Increasing *Tct* will improve the required degree of consensus and security for transaction completion. However, if the node is unable to send confirmation messages in unstable network conditions, it will cause network congestion and eventually fail to reach consensus. Declining *Tct* reduces the number of validators needed to confirm fraudulent transactions. Even if the fraudulent transaction is challenged, it will harm the alliance chain throughput, communication, and storage. The dynamically adjusted *Tct* can balance the performance, security, and stability of the alliance chain, providing flexibility to meet different application scenarios.

The modification of *Tct* by the Leader node will affect the alliance chain's attributes, which gives the Leader node higher authority than the consensus node. The reputation model has no direct precautions to prevent the leader node from changing *Tct* in reverse.

The protocol sets *Tct* to be adjustable only once during the tenure of any Leader node, and there is a configurable range for this adjustment. An attacker would require multiple consecutive malicious Leader nodes to impact the system negatively, thus increasing the cost of malicious operations.

## 4. Correctness Argument

### 4.1. Performance Analysis

There are $\frac{n}{k}$ nodes in the shard, the time for a node to process a message is fixed at $t_1$, and the message delivery time is fixed at $t_2$. $T$ is the time required to complete the intra-shard consensus process in this protocol. According to the dynamic transaction confirmation consensus process, there are four stages of message transmission. Nodes must process a *Tct* message, a transaction message, and *Tct* confirmation messages. The time complexity for achieving consensus is calculated in Equation (4).

$$T = 4t_2 + (Tct + 2)t_1 \tag{4}$$

Set $T'$ as the time required to complete consensus in the PBFT. The PBFT consensus process has a total of 5 stages of message transmission. Nodes in a shard need to process a request message, a pre-prepare message, $\frac{2n}{3k}$ prepare messages, and $\frac{2n}{3k}+1$ commit messages. The time complexity of achieving consensus is calculated as shown in Equation (5).

$$T' = 5t_2 + (\frac{4n}{3k} + 3)t_1 \tag{5}$$

Equation (6) is obtained from (4) and (5).

$$T' - T = t_2 + (\frac{4n}{3k} + 1 - Tct)t_1 \tag{6}$$

In the intra-shard transaction consensus phase, the dynamic transaction confirmation consensus outperforms PBFT in terms of efficiency. The specific value of the improvement is jointly determined by $t_1$, $t_2$, $\frac{n}{k}$, and *Tct*.

The values of *Tct* are set to $Tct_1$ and $Tct_2$, and the difference in message confirmation completion time is equal to the reduced transaction confirmation latency $\triangle T$ as shown in Equation (7).

$$\triangle T = |Tct_1 - Tct_2|t_1 \tag{7}$$

If the *Tct* in the dynamic transaction confirmation consensus changes, the transaction latency will also change. $t_1$ is usually measured in milliseconds, so *Tct* has little impact on system throughput. $T_0$ is the time for the consensus committee to reach a PBFT consensus after a Leader node submits the microblock. The number of shards affects both the microblock verification and bigblock packaging time, as shown in Equation (8).

$$T_0 = 5t_2 + (\frac{4k}{3} + 3)t_1 \tag{8}$$

### 4.2. Stability and Safety

In this protocol, the Leader node broadcasts *Tct* messages (a total of $\frac{n}{k} - 1$ messages) in the shard. The client segment sends transaction information to each node in the shard (a total of $\frac{n}{k}$ messages), and each node will broadcast a confirmation message (a total of $\frac{n}{k} * [\frac{n}{k} - 1]$ messages). The number of messages $S$ required to complete transaction confirmation in a shard is shown in Equation (9).

$$S = (\frac{n}{k} - 1)^2 + \frac{3n}{k} - 2 \tag{9}$$

According to the PBFT process, the client sends a request message to the Leader node, which broadcasts a pre-prepare message to each consensus node (total of $\frac{n}{k} - 1$ messages). Each node broadcasts a prepare message (total of $[\frac{n}{k} - 1]^2$ messages) and a commit message (total of $\frac{n}{k} * [\frac{n}{k} - 1]$ messages). The number of messages required to achieve consensus in a shard with the same configuration is shown in Equation (10).

$$S' = 2(\frac{n}{k} - 1)^2 + \frac{2n}{k} - 1 \qquad (10)$$

The intra-shard consensus of the protocol requires only about half the number of messages required by PBFT to achieve consensus. Under the same computer and network conditions, the consensus of this protocol is less negatively affected by the increase of nodes than PBFT. This protocol has better stability and can also reduce the consumption of network resources and storage space. If multiple verifications of the system are due to mishandling of transactions by validator nodes or wrong transaction messages, the alliance chain will not return to normal until the malicious nodes are removed. Malicious verification nodes or wrong transactions can cause *Tct* to affect stability.

The existing protocols for selecting a Master node have certain characteristics. One approach involves selecting a different node as a Master node in each epoch or round according to the rotation rules, which introduces latency and additional communication overhead and allows an attacker to control the rotation order [7,35]. Another approach is staking-based selections, which incentivize nodes to follow the rules and maintain normal behavior [47]. Over time this could lead to centralization within the system. Performance-based selection can improve system efficiency but can cause some nodes to become masters, continually reducing the utilization of others [34]. The protocol design utilizes locally generated verifiable and unpredictable random values for selections. This approach has several advantages, such as protection against attacks and falsification of lottery result, and maintaining low energy consumption and uniqueness of result.

Regarding node assignment, methods based on node properties (such as liveness and performance) have centralization risks [22,35], while free-choice assignment methods are less resistant to malicious behavior [28]. In the protocol design, the identification ID for node assignment calculation considers how nodes can manipulate their IP addresses using proxies or other methods. The node assignment is achieved through cryptographic calculations of various parameters, incorporating the public key and the epochRandomness involving the participation of multiple nodes. While not eliminating the possibility of manipulating individual parameters, this guarantees a robust and manipulation-resistant identity generation process.

Consensus nodes are responsible for the duties of validating transactions and sending challenge messages. Both wrong verification and challenge failure will reduce the reputation of the consensus node. Leader node is responsible for adjusting the transaction confirmation threshold, counting the number of confirmation messages, and generating microblocks. If the Leader node sends different values of *Tct* to each consensus node or does not send to some consensus nodes, the consensus node will replace the Leader node due to inconsistent status. If the consensus node finds that the *Tct* and voting information in the microblock are wrong, it will consider the Leader node malicious and send a challenge message to the Master node. The access mechanism of the alliance chain combined with the adjustment upper limit of *Tct* makes it very expensive for the Leader node to adjust *Tct* reversely. Master node have the responsibility to process challenge messages. If the Master node does not process the challenge message in time, the consensus node will broadcast the proposal to replace the Master node to all nodes. After the Master node is replaced, the new Master node will process unprocessed challenge messages. It can be concluded from the above that the protocol is safe and practical.

## 5. Experimental Design

The experiment involves comparing the performance of the sharding protocol with mainstream sharding protocols in the alliance chain and testing the impact of the transaction confirmation threshold and block size on system performance. The simulated alliance chain has a consistent system architecture and network model, and any differences are controlled within the scope of the protocol as much as possible.

### 5.1. Experiment and Configuration

The clients and nodes are simulated using a simulation system written in C++ and utilizing multi-threading technology. Transactions are transmitted by the clients in this test, and each shard consists of multiple consensus nodes and a Leader node. The system is divided into the transaction module and the consensus module. The system's performance and scalability will be evaluated based on its throughput and transaction latency. The detailed configuration is shown in Table 1.

**Table 1.** Software and hardware environment configuration.

| Software and Hardware Environment | Configure |
| :---: | :---: |
| CPU | 2.40 GHz Intel Core i5-9300H |
| RAM | 16 GB 2667 MHz DDR4 |
| System | Windows 11 |

### 5.2. Experimental Testing

The performance compares the throughput and transaction confirmation latency of different sharding protocols for one epoch in the alliance chain with a shard configuration of 4 or 6 nodes. The performance expands the alliance chain by increasing the number of shards from 4 to 8 to evaluate the impact of different network sizes on protocol performance.

According to the data in Figures 6a and 7a, this protocol has a throughput difference of more than 66% compared to traditional Elastico. Compared to Sharper and AHL sharding protocols for permissioned chain, this protocol increases throughput by 20% and 27%, respectively. Compared with high-performance Fleetchain, the throughput of this protocol has increased by 19%. With the expansion of network scale, this protocol maintains performance advantages compared with other sharding protocols, highlighting its superior scalability. Figures 6b and 7b describe the transaction confirmation latency comparison between this protocol and mainstream sharding protocols as the number of nodes in each shard increases. Although the performance gap between this protocol and Fleetchain has narrowed, it still achieves a 12% reduction in latency over the latter. An overview of the above results shows that the system performance advantage of this protocol remains stable as the number of shards increases.

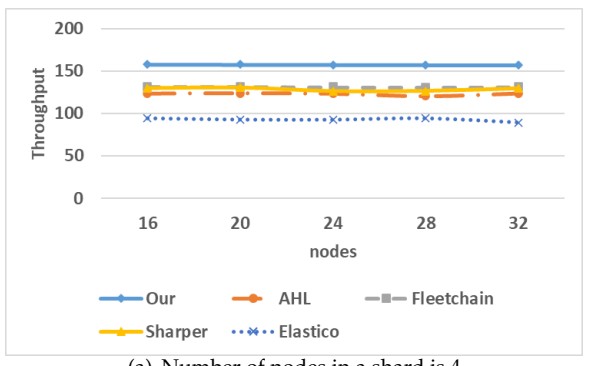

(a) Number of nodes in a shard is 4.

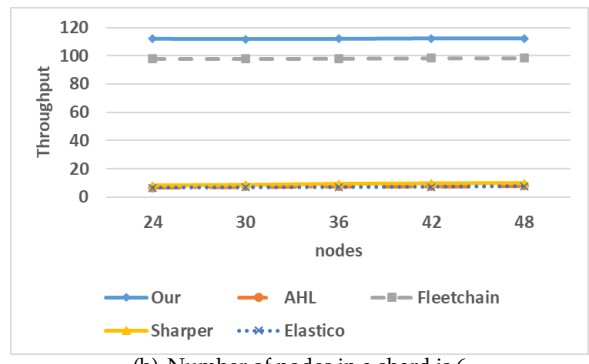

(b) Number of nodes in a shard is 6.

**Figure 6.** System throughput averages comparison for shard counts ranging from {4, 5, . . . , 8}. Other factors (*Tct* = 3, block size = 800 transactions) are the same except for the sharding configuration.

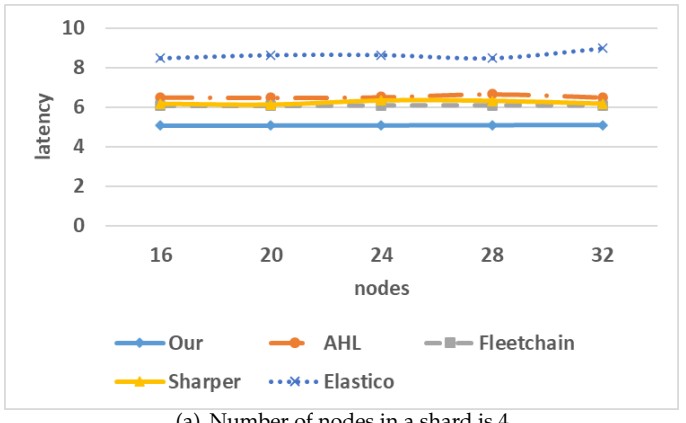
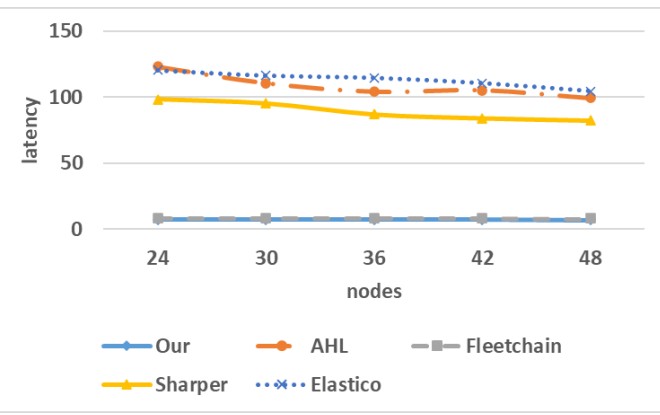

(a) Number of nodes in a shard is 4.          (b) Number of nodes in a shard is 6.

**Figure 7.** System latency averages comparison for shard counts ranging from {4, 5, . . . , 8}. Other factors (*Tct* = 3, block size = 800 transactions) are the same except for the sharding configuration.

Next, test whether *Tct* affects system performance. Set up a client to send transactions to alliance chain with different shard configurations (4 shards, each with 8 or 9 nodes) and adjust *Tct* simultaneously. Measure the performance of the alliance chain under different *Tct* and average the results, as shown in the Figures 8 and 9.

Experimental results show that *Tct* does not affect the system throughput or transaction confirmation latency in the absence of erroneous transactions. Therefore, at the initial stage of system operation, *Tct* should be set within the range of $\frac{1}{3}$ to $\frac{2}{3}$ of the number of shard nodes to ensure stability.

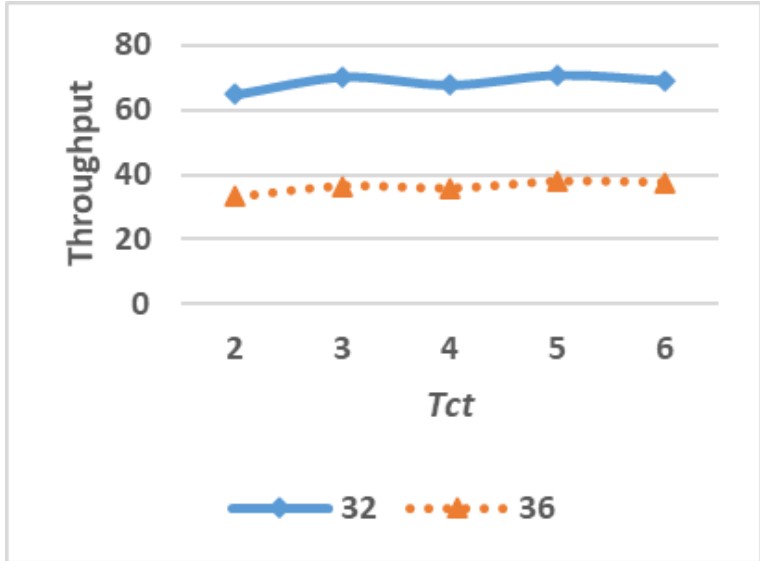

**Figure 8.** System throughput averages comparison for different *Tct* values from {2, 3, . . . , 6}. Experiments are conducted in two shard configurations, where the number of shards is 4, but the number of nodes within the shards is 8 and 9 (the total number of nodes is 32 and 36), respectively.

This experiment tests the effect of the total amount of transactions in an epoch on the performance of the protocol. Control the shard configuration and *Tct*, only adjust the size of the bigblock, and take the average value of the performance of multiple measurement systems, as shown in Figure 10.

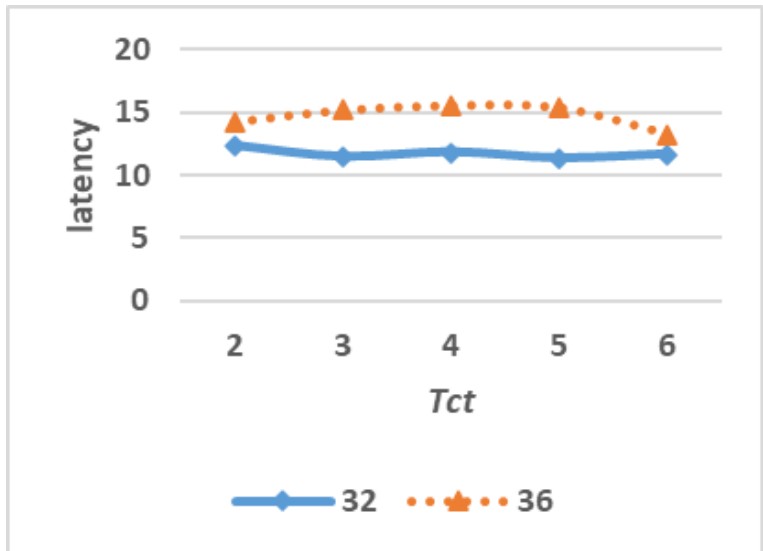

**Figure 9.** System latency averages comparison for different *Tct* values from {2, 3, . . . , 6}. Experiments are conducted in two shard configurations, where the number of shards is 4, but the number of nodes within the shards is 8 and 9 (the total number of nodes is 32 and 36), respectively.

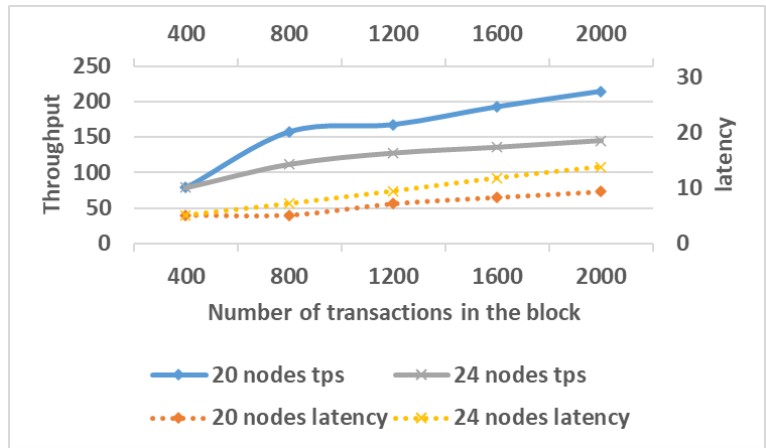

**Figure 10.** Performance averages of the system for different block sizes from {400, 800, . . . , 2000}. Experiments are conducted in two shard configurations, where the number of shards is 4, but the number of nodes within the shards is 5 and 6 (the total number of nodes is 20 and 24), respectively.

Increasing the block size can achieve the goal of enhancing the throughput but will result in longer block times. Therefore, when setting the block size in practice, it is necessary to consider the requirements of the application scenario for throughput and transaction confirmation latency.

While the simulations may not perfectly reflect the complexities of real distributed environments, they allow researchers to examine and evaluate the protocol's behavior under controlled conditions. The experiments aim to understand the overall process and evaluate the scalability improvement provided by the protocol compared to existing methods.

Additionally, there may be differences when applying the protocol to a real distributed environment. It is important to note that these differences do not hinder the implementation of the protocol process in a distributed environment or undermine the potential for enhanced scalability and advantages over other protocols.

## 6. Conclusions

This paper proposes a dynamic transaction confirmation consensus sharding protocol specially designed for alliance chain systems. The protocol implements dynamic trans-

action confirmation consensus as an intra-shard consensus and uses parallel processing of microblocks and submission of sequential proposals on different shards to achieve this. In addition, the protocol also includes node assignment, review mechanism, and reputation model to prevent attackers from centralizing their controlled nodes into shards, thereby reducing system latency and duplicate message propagation caused by attacks. Experimental results show that the proposed sharding protocol prominently improves the scalability of alliance chain compared to other methods. In order to ensure that users can frequently access the alliance chain and use a considerable number of non-faulty nodes under a smooth network environment, it is necessary to adjust the dynamic confirmation threshold and block capacity. In addition, the protocol can also be applied to networks with the subpar performance of node facilities.

This protocol utilizes epochRandomness in calculating the identity ID to prevent high-computing power nodes from monopolizing the position of the Leader node for an extended period. This approach consumes much computing power to establish identities instead of verifying and reviewing transactions. To balance system resource consumption and ensure fairness, future work considers extending the epoch cycle, increasing the block capacity, and adopting different selection methods to facilitate node role rotation when appropriate. By improving the practicability of the protocol, researchers can expand the application scenarios from alliance chain to various blockchains.

**Author Contributions:** Conceptualization, N.S. and J.L.; methodology, N.S.; validation, N.S. and J.L.; formal analysis, N.S., J.L. and Y.Z.; investigation, J.L.; resources, N.S.; data curation, J.L.; writing—original draft preparation, J.L.; writing—review and editing, N.S. and Y.Z.; visualization, J.L.; supervision, N.S.; project administration, N.S.; funding acquisition, N.S. All authors have read and agreed to the published version of the manuscript.

**Funding:** This research was funded by Postgraduate Research & Practice Innovation Program of Jiangsu Province: KYCX22_3059.

**Institutional Review Board Statement:** Not applicable.

**Informed Consent Statement:** Not applicable.

**Data Availability Statement:** Not applicable.

**Acknowledgments:** The authors would like to thank Postgraduate Research & Practice Innovation Program of Jiangsu Province for their financial support.

**Conflicts of Interest:** The authors declare no conflict of interest.

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
