# Peer review of "Dynamic Transaction Confirmation Sharding Protocol for Alliance Chain"

_applsci, doi:10.3390/app13126911_

Round 1

Reviewer 1 Report

The research paper introduces a novel dynamic transaction confirmation sharding protocol for alliance chains. Current implementations face scalability challenges when the number of nodes increases, leading to communication congestion and throughput drop. To address these issues, the paper proposes a protocol that enhances transaction processing efficiency by dividing nodes into shards and assigning different transactions to different shards. The protocol employs dynamic transaction confirmation consensus as a sharding intra-consensus mechanism, minimizing message size and packaging transactions into microblocks. This approach modifies communication content during transaction propagation among shards, reducing network congestion and shard reconfigure cost.

Simulation results demonstrate that the proposed protocol outperforms existing permissioned chain sharding protocols. The protocol is particularly suitable for industrial and commercial applications requiring high throughput and reliability in fields such as finance, logistics, and supply chain management.

It can be stated that the paper is well-written and provides a clear overview of the proposed protocol and its advantages. However, it could be beneficial to include more details about the C++ implementation, particularly specific descriptions of the algorithms employed. This addition would enhance the paper's technical content and make it more comprehensive for readers interested in the implementation aspect. How well the protocol would run in a distributed environment (not simulated) is also unclear.

Author Response

Dear reviewer,

Many thanks for your constructive remarks. We adapted our manuscript, to answer your concerns. All changes are marked in track changes in the document attached. Note that we also received feedback from other reviewers, which is also reflected in these changes in the manuscript.

Best wishes,

The authors

Reviewer 2 Report

I consider that I am not an expert in this domain, but I would like to know if the number of nodes in a real environment varies between 16 and 48. This variation seems too limited to carry out an experiment that validates scalability. I would expect more exponential scaling for the number of nodes.

Author Response

Dear reviewer,
Many thanks for your constructive remarks. We have answered your concerns in the attached document. Note that we also received feedback from other reviewers, which is also reflected in these changes in the manuscript.
Best wishes,
The authors

Reviewer 3 Report

The paper proposes a protocol for improving the performance of alliance chain by enhancing the efficiency of shard consensus mechanism. The paper needs revisions in terms of:

1- English language: for example line 33 and others 

2- spaces between words and reference number

3- citing more reference, for example, the limitation of blockchain, limitations of on-chain and off-chain

4- the introduction does not show clearly the relation between shards and alliance chain

5- a subsection of alliance chain should be added to the related work to explain what is alliance chain, where it is used and its limitations -- adding more references too

6- adding more information to figure captions 

7- equation 3 should be explained more, giving example about the threshold node reputation value

8- how the randomly selected master node is guaranteed to achieve stability and safety, discussion of other mechanism to select master nodes and node assignment is needed 

9- adding future work to the conclusion section by highlighting this work limitations and other unresolved issues

1- English language should be revised to fix some minor errors: for example line 33 and others 

Author Response

(The authors gave the same response as above.)

Round 2

Reviewer 3 Report

The authors made the corrections and addressed my comments